# Cocaine-Induced Midline Destructive Lesions: A Real Challenge in Oral Rehabilitation

**DOI:** 10.3390/ijerph18063219

**Published:** 2021-03-20

**Authors:** Andrea Rampi, Alessandro Vinciguerra, Stefano Bondi, Nicoletta Stella Policaro, Giorgio Gastaldi

**Affiliations:** 1Division of Head and Neck Department, Otorhinolaryngology Unit, IRCCS San Raffaele Scientific Institute, 20132 Milan, Italy; rampi.andrea@hsr.it (A.R.); bondi.stefano@hsr.it (S.B.); 2School of Medicine, Vita-Salute San Raffaele University, 20132 Milan, Italy; 3Dental School, Vita-Salute San Raffaele University, 20132 Milan, Italy; stellapolicaro@hotmail.it (N.S.P.); gastaldi.giorgio@hsr.it (G.G.); 4Department of Dentistry, IRCCS San Raffaele Scientific Institute, 20132 Milan, Italy

**Keywords:** cocaine-induced midline destructive lesions, CIMDL, reconstructive surgery, palatal perforations, prosthetic rehabilitation, palatal obturators

## Abstract

Cocaine abuse is associated with severe local effects on mucosal and osteocartilaginous structures, with a centrifugal spreading pattern from the nose, a condition known as cocaine-induced midline destructive lesions (CIMDL). When the soft or hard palate is affected, a perforation may occur, with subsequent oro-nasal reflux and hypernasal speech. Both diagnosis and therapy (surgical or prosthetic) constitute a serious challenge for the physician. The cases of three patients affected by cocaine-induced palatal perforation and treated with a palatal obturator at San Raffaele Dentistry department between 2016 and 2019 are presented. In addition, the literature was reviewed in search of papers reporting the therapeutic management in patients affected by cocaine-induced palatal perforation. All the patients in our sample suffered from oro-nasal reflux and hypernasal speech, and reported a significant impact on interpersonal relationships. The results at the delivery of the obturator were satisfactory, but the duration of such results was limited in two cases, as the progression of the disease necessitated continuous modifications of the product, with a consequent increase in costs and a reduction in patient satisfaction. In conclusion, the therapy for palatal defects in CIMDL includes both reconstructive surgery and prosthetic obturators, the latter being the only possibility in the event of active disease. It successfully relieves symptoms, but the long-term efficacy is strongly related to the level of disease activity.

## 1. Introduction

Cocaine abuse is a serious and increasing problem for public health care in Europe [1]. Its systemic absorption is responsible for immediate and long-latency diseases, which can involve any system or organ, especially cardiovascular and psychiatric alterations [2]. The highest concentration is reached at the site of absorption and, as sniffing is the most common way of consumption, sinonasal mucosa is frequently involved [1]. The most acknowledged pathophysiologic mechanism is the drug-induced ischaemia of the tissues, but impaired mucociliary clearance, superinfections, and autoimmunity are other important harmful mechanisms [3,4,5,6]. The intensity of the damage is extremely variable and only partially related to cocaine intake: it may be limited to the mucosa, or go deeper and, usually having a centrifugal pattern of distribution, progressively erode the septum, turbinates and lateral and inferior walls of the nose, or involve the external region of the philtrum and columella [3,4,5,6,7]. These lesions are described as cocaine-induced midline destructive lesions (CIMDL) and are often responsible for a significant reduction in patients’ quality of life [3,8]. Depending on their extension, in fact, they can give rise to nasal obstruction, hyposmia, epistaxis, severe facial pain, and aesthetic nasal deformities [3,5,7,8,9]. When soft or hard palate perforations occur, patients experience hypernasal speech and oro-nasal reflux, with a consequent serious negative impact on social life [5,10,11].

Unfortunately, no test can definitely attribute a mucosal destructive lesion to cocaine consumption, and cocaine abusers often deny their addiction [12]. As a consequence, many diseases must be considered as possible differential diagnoses for CIMDL, including vasculitides, aggressive infections, tumors, and trauma, and a diagnosis of CIMDL is often based on the exclusion of these alternatives [3,5,8,13,14].

Palatal perforations can be treated with palatal obturators or surgery, but the latter option is doomed to fail if drug consumption persists. Unfortunately, reports on the treatment of palatal defects induced by cocaine are limited, and even more inadequate is the literature on the manufacturing process and post-delivery care of palatal obturators. Therefore, in the present paper, we reassess the prosthetic management and follow-up of three patients treated with a palatal obturator between 2016 and 2019 at San Raffaele Dentistry department, comparing our experience with other reports in the literature.

## 2. Materials and Methods

### 2.1. Diagnosis

At San Raffaele hospital, patients affected by palatal perforations of undetermined aetiology and attending a Dental or Otolaryngologist outpatient clinic are managed through a multidisciplinary agreement between an ENT specialist, an oral surgeon, and other specialists (radiologists, rheumatologists etc.), depending on the clinical presentation. The diagnostic algorithm includes a clinical examination of the oral and nasal cavity (through an endoscope), appropriate imaging (CT or MRI), biopsies, and blood tests. Urine or hair tests are often required to diagnose cocaine addiction, but are unnecessary if the patient admits the habit, as in the present sample. In this study, in fact, we retrospectively reviewed the cases of three patients affected by cocaine-induced palatal defects and treated with a prosthesis between 2016 and 2019, who confessed their present or past addiction. The present study was approved by the San Raffaele Hospital ethics committee in adherence with the Declaration of Helsinki (15 July 2020).

### 2.2. Preliminary Evaluation and Production of the Prosthesis

After the palatal lesion was diagnosed as cocaine-related, patients were strongly advised to follow a rehabilitation program with a dedicated structure and were evaluated for a prosthetic approach to reduce their symptoms. Clinical examination and an orthopantomography were performed to assess the condition of residual dentition, and the adequacy of eventual previous obturators was rated. If the previous treatment was considered adequate, adjustments were made to adapt it to the new oral condition; otherwise, a new prosthesis was made. In the latter case, alginate impressions were taken, and on their base, a study model made of hard plaster was created. The study model was used to produce an individual resin tray to take a definitive impression using medium intensity polysulphides. The latter material was chosen for its ability to identify conflict areas with oral soft tissues (mouth muscles, fraenum, etc.); a plaster master model was then designed with some grooves, which prevented such soft tissues from displacing the obturator. In this way, the master model was used to produce a removable prosthesis, consisting of a chrome–cobalt alloy framework coated with acrylic resin, with a superior bulb to seal the defect. As the residual dentition in all three patients was considered sufficient for adequate stabilization of the prosthesis, the product was provided with hooks to anchor it to the dental elements (Figure 1).

### 2.3. Post-Delivery Care

At the moment of delivery, the stability of the product and the effects on deglutition and phonation were assessed, patients were instructed on the correct use and cleaning of the prosthesis, and a close follow-up was scheduled. The frequency of visits depended on disease activity and could vary from once a week to once a month. If the patient suffered from persistent tissue inflammation, increase in the size of the lesion, or decubitus, the prosthesis was relined directly in the mouth with temporary autopolymerising acrylic resin and then polished; close visits were consequently necessary. On the other hand, if the patient refrained from drug use and the lesion remained stable, the reline could be made with autopolymerising acrylic resin with reduced contraction, which is a more stable material, and the follow-up was therefore only once a month. At every visit, prosthetic stability previous to and after the reline was assessed and compared to other visits, and the satisfaction of the patient was recorded.

## 3. Results

### 3.1. Main Findings

The three patients in our sample were a 47-year-old woman and two men, ages 49 and 54. At the first visit, the oro-nasal communication involved the hard palate in one case, the soft palate in another case, while one patient suffered a hard and soft palate erosion. When nasal endoscopy was performed, all patients were found to have a septal perforation and destruction of at least one turbinate, in addition to mucosal inflammation and crusting. These findings were confirmed by CT scan, which identified a typical centrifugal spreading of lesions. Cultures from samples taken during endonasal endoscopy were positive for *Staphylococcus aureus*, while biopsies showed a non-specific polyclonal plasma cell component in all cases.

### 3.2. Symptoms and Post-Delivery Care 

When asked about their symptoms, all patients revealed the defect had a strong impact on their social and work life, due to difficulties in speech, nutrition, and swallowing. The symptoms were related to both the dimensional extension of the defect and its activity, and increased in the event of progressive tissue erosion.

At the moment of delivery of the prosthesis, all patients showed a significant reduction in symptoms and were satisfied with the product. Analogously, any reline received positive immediate feedback. Nevertheless, the results deteriorated rapidly for the two patients who had active disease. In the event of continuous inflammation, in fact, the oral tissues underwent progressive erosion, which rendered the prosthesis inappropriate for the patient and led to the need for close follow-up and continuous relining, with a consequent increase in costs and a reduction in patient satisfaction. One of the two patients who had disease progression admitted to still consuming cocaine, and the other, despite denial, was suspected of analogous behavior.

### 3.3. Patient Presentation 

Patient 1: At the first visit, the palatal lesion was limited (approximately 1 cm in diameter), but it was sufficient to cause phonation and swallowing difficulties. Thus, progressive disease was observed, and the first obturator, despite frequent relining, had to be replaced after 3 years with a new product, with further relining needed in the following year.

Patient 2: This patient constitutes the most successful outcome in the sample. At the first visit, he reported severe oro-nasal reflux and serious phonation and a swallowing deficit caused by a large hard palate perforation. He was strongly motivated to refrain from cocaine use, and the lesion remained stable and without evident inflammation on the borders. As a consequence, the removable obturator had good long-term efficacy on symptoms and only minimal refining of the shape was necessary in 18 months. The patient reported satisfaction for the product at all times during follow-up, which is still active. Figure 2 displays the perfectly shaped prosthesis on a coronal CT scan.

Patient 3: The patient suffered from a hard and soft palatal perforation, which had already been treated with a palatal obturator (Figure 3a). Nevertheless, it had lost efficacy due to the increase in the size of the lesion. The patient complained about difficulties in deglutition and a hypernasal voice that made his speech almost unintelligible, but the prosthesis was considered congruous. Nevertheless, the relines of the product became more and more frequent as a consequence of disease activity, which was attributable to persistent drug use (admitted by the patient) during a 3-year follow-up. In the end, the lesion reached an incredibly large size, with a massive centrifugal erosion of the midline structures, as displayed in Figure 3b. At present, he still reports severe symptoms and an unsatisfactory result of the prosthesis.

## 4. Discussion

Cocaine is a widely used stimulant drug, with a broad spectrum of local and systemic effects [5]. Its local impact may induce, in a limited percentage of cases, CIMDL [5,15]. Among the effects, the most frequent is septal perforation, but the erosive process, usually following a centrifugal direction, may affect the turbinates, lateral nasal walls, orbits and the clivus, or extend to the face with disfiguring lesions of the columella, superior lip, and maxilla [5,6,7,8,14,16,17,18]. When the nasal floor is involved, an oro-nasal communication may be responsible for food reflux from the nose, hypernasal speech, and, especially for soft palate defects, deglutition difficulties [10]. CIMDL constitute a challenge for the physician from both diagnostic and therapeutic points of view. This is further exacerbated by the scarcity of data in the literature on the topic. The present article was therefore aimed at presenting the experience at our center in the diagnosis, management, and therapy of cocaine-induced palatal perforations, as an aid for any physician having to treat this condition.

Ascribing a palatal perforation to cocaine consumption may be, in fact, challenging: patients often deny drug abuse and many differential diagnoses must be considered [3,9,18,19]. The differentials should include tumors (squamous cell carcinoma, melanoma, minor salivary gland and lymphoreticular neoplasms, metastases), trauma, infections (syphilis, tuberculosis, mycotic infections, etc.), and immunity disorders (especially granulomatosis with polyangiitis) [4,5,12,14,16,18,20,21,22,23]. Some are easily distinguishable through anamnesis, blood tests, or biopsies, while granulomatosis with polyangiitis (GPA) constitutes a more difficult differential diagnosis, as the histologic findings and antineutrophil cytoplasmic antibody (ANCA) positivity often have common features, although a GPA limited to the upper airways is uncommon [3,5,12,21,24,25,26]. In this regard, Wiesner et al. [24] found an ANCA perinuclear variant that binds the human neutrophil elastase (HNE) to be common in CIMDL and rare in autoimmune diseases, suggesting the utility of these antibodies for differential diagnosis. In addition, Trimarchi et al. [8] found single multinucleated giant cells or granulomas and microscopic foci of deeply located necrosis to be pathognomonic histologic findings for GPA. Despite all these clues, however, a diagnosis of CIMDL is often difficult without an admission of cocaine abuse by the patient.

If the diagnosis of CIMDL still has obscure aspects, there is even more uncertainty related to treatment: given the scarcity of reports in the literature, there is no solid evidence for effective therapy [3,5]. On the basis of a suspected imbalance in immunity, immunosuppressive agents have been proposed by some authors [27,28], but others do not recommend this therapeutic approach [3,5,29]. Therefore, a primary effort in the treatment of the disease must be given to counselling, and patients must be encouraged to abstain from drugs, as it is the only effective strategy to stop the progression of the disease [30]. The authors suggest that the patient should follow a drug rehabilitation program with a dedicated and professional structure to increase the possibility of success, as persistent drug abuse strongly impacts any therapeutic approach.

Focusing on the management of palatal perforation, different strategies have been reported, but the scarcity of data in the literature makes it difficult to compare their outcomes. The two possible approaches are reconstructive surgery and palatal prosthetics; the first option has the advantage of being a definitive solution, which has a limited need for successive reevaluation and can be performed with local, regional, or distant flaps [2,4,9,31]. Local flaps are harvested in a similar way to reconstructive surgery for other causes, but, in contrast, the tissues are often frail due to cocaine-induced ischaemic damage and cannot be sutured under tension [2,31]. As a consequence, this surgery, which includes reverse mucosal flaps, rotational flaps, and bilateral pedicled mucoperiosteal flaps, is not suitable for large defects or a high palatal vault [2]. Regional flaps include the temporalis muscle flap and the dorsal pedicled tongue flap [2,31]. This approach allows wide defects to be covered, since the vascular pedicle is preserved; on the other side, the temporalis muscle flap necessitates a tunnel in the maxilla and can lead to aesthetic alterations, and the dorsal pedicled tongue flap obliges intermaxillary fixation for 3 weeks followed by a second-stage surgery to sever the flap from the tongue [2,32]. In the end, distant flaps, and more commonly a radial forearm free flap, can be used to close the defect; they can cover wide areas, but they require a skin graft to cover the donor site, where resulting aesthetic defects are relevant [2]. With regards to outcomes, there are many reports of successful surgery and long-lasting resolution of oro-nasal communications, but they refer to a selected and well-motivated sample [2,11,31,32]. Performing surgery on a patient still using cocaine would determine immediate failure, and for this reason many surgeons require a period of 6 to 18 months of abstinence before considering the surgical option [2,7,11,18,19,32,33].

Prosthetic management constitutes another valid option, and may be intended both as a definitive solution and as a bridging measure to surgery [9]. Obturators are usually made of a metal framework and an acrylic resin structure which covers the fistula, eventually provided with a bulb or a pharyngeal extension [9,10,34]. The key-point for their success is reliable stability, which can be provided through adhesives, through hooks secured to teeth or, for edentulous patients, through specific implants [9,10,34,35,36,37,38,39,40]. The main advantage of a prosthetic is that an obturator can also be used in the presence of active disease, as relining can be performed to adapt the prosthesis to a widening lesion [2,9]. On the other hand, frequent follow-up is necessary, and some patients complain of discomfort in the use of the product [9,34]. Unfortunately, reports describing the manufacturing technique of obturators are rare, and reports on the post-delivery care are even rarer; for this reason, we believe the present paper can help in raising awareness about the topic and its management.

In our experience, two patients experienced active disease for the entire period of follow-up, while the third always had stable disease and reported complete satisfaction with his obturator. Consequently, none were considered for surgery. The adhesive option was considered insufficient for a reliable stability and, as all these patients had at least partial dentition, they were treated with a removable obturator secured to the teeth. Regarding the results, we believe that the prosthetic rehabilitation in CIMDL must not be evaluated according to the reduction of symptoms at the moment of the delivery or after a reline, as this target was achieved in all cases. The index of a successful rehabilitation must be the temporal extension of the prosthetic benefit: frequent relines are expensive and time-consuming, and they are responsible for a reduction in the patient’s quality of life and satisfaction with the product [9]. This assumption is confirmed by the present sample, as the two patients with active disease had reduced satisfaction for their therapeutic management compared with the third one. A possible explanation for the activity of the disease is the persistence of drug use, which was admitted in one case and suspected in the other one: a fact that confirms the critical need for a drug rehabilitation program in parallel with prosthetic therapy.

As in any disease, knowledge of the pathogenic actors involved are crucial for correct management, and the factors which determine the progression of CIMDL are presumably the same as those that determine their occurrence. Unfortunately, cocaine is believed to give rise to CIMDL through several different pathogenic pathways, and only some are known [5]. Firstly, the drug induces an immediate and short-latency mucosal ischaemia; its vasoconstrictive properties and its enhancement of platelet aggregation and decrease in fibrinolytic activity may then lead to permanent vascular injury after repeated insufflation [22,41,42]. Such events can determine necrosis in oro-nasal mucosa, and it must be highlighted that the reduced perfusion of the surrounding still-viable tissues is responsible for a limited resilience to additional harms, including the persistence of cocaine consumption, infections, and reconstructive surgery [2,5,11,42]. Moreover, the frequent association of cocaine addiction and tobacco consumption can determine an additional reduction in tissue perfusion [42]; consequently, refraining from tobacco consumption is advised. Cocaine also paralyzes mucosal cilia, and impaired mucociliary clearance is responsible for chronic inflammation and scabs, which increase the discomfort and symptoms of patients [22,43]. In addition, its crystals and its adulterants cause direct mechanical aggression to the nasal mucosal lining, but this effect is often underestimated by the patient due to the drug anaesthetic effect [5,22,29,41]. As a consequence of the altered physiology and immunobalance of endonasal structures, bacterial infections become more probable, and *S. aureus* has been reported in many publications [8,28,33,44]. An autoimmune response has also been recently thought to play a key role in CIMDL, especially as ANCA has been found in a consistent percentage of patients affected [3,24,25]. Cocaine-induced autoimmune disease would explain the possibility of a progressive erosion even after prolonged abstinence, the high percentage of women affected, and the histologic similarities with some autoimmune disorders [3,6,8,12,24,25,26]. Nevertheless, the triggering factor for the autoimmune response in CIMDL is still discussed. It may be related to the production of bacterial superantigens as a consequence of the previously mentioned superinfections, or it may be due to polyclonal B-cell stimulation induced by cocaine or drug adulterants [3,28,29]. Some authors, in fact, have specifically highlighted the property of adulterants, and levamisole in particular, in dysregulating immunity, suggesting the existence of an autoimmune spectrum induced by different substances in predisposed patients [29,45]. Additional support for the role of autoimmunity is given by the evidence that ANCA are the distinctive feature between CIMDL and non-CIMDL drug users [24]. The reason for this finding is still unclear, but the role of ANCA may depend on the direct induction of proteolytic activity, the enhancement of the inflammatory response to injury, or on the opsonization of pre-apoptotic cells and subsequent increase in inflammatory cytokines [5,24,25,46]. The apoptosis pathway, in fact, has also been suggested as an additional player in CIMDL, as Trimarchi et al. demonstrated dose-dependent expression of apoptotic markers (Caspase-3 and Caspase-9) in nasal mucosal cells from cocaine abusers [22].

For the abovementioned reasons, when choosing the proper therapeutic approach, the damage that the tissues can withstand before necrosis must be considered according to the margin conditions, the persistence of drug abuse, the presence of other harmful mechanisms, and the personal characteristics of the patients (i.e., infections, tobacco use, patient age, vascular diseases etc.).

## 5. Conclusions

The treatment options for a palatal perforation induced by cocaine are reconstructive surgery, which is possible only in the event of stable disease, and a prosthetic. The latter option is valuable in the event of stable disease, for which it can be considered the first treatment option. It is also important in the active form of the disease, but the results in these cases are much less satisfactory.

## Figures and Tables

**Figure 1 ijerph-18-03219-f001:**
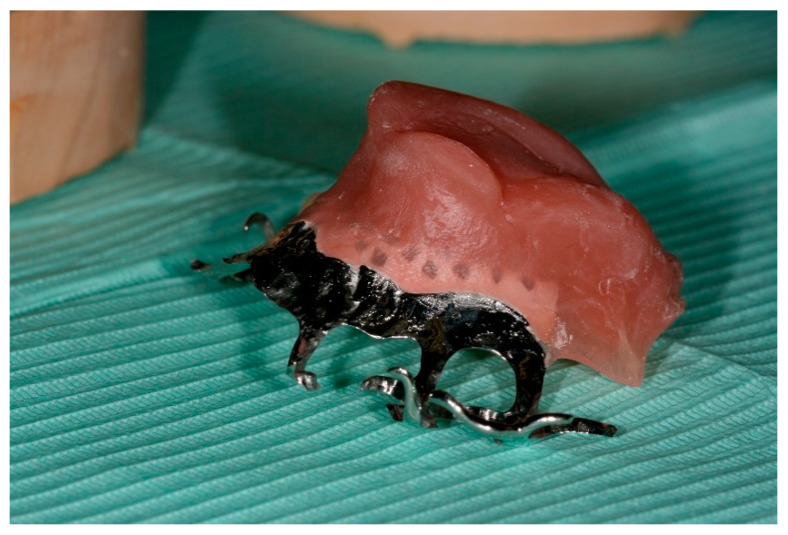
A palatal obturator.

**Figure 2 ijerph-18-03219-f002:**
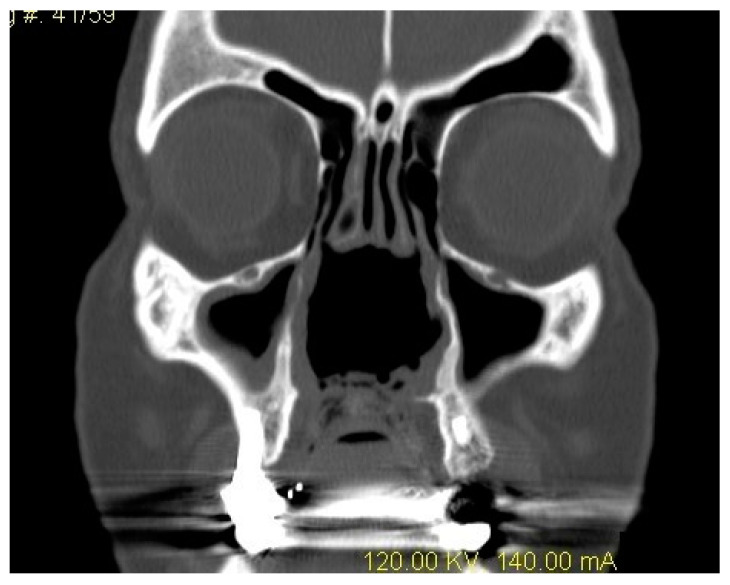
Patient 2: A coronal CT scan showing a wide erosion of midline structures (septum and turbinates), including a palatal perforation treated with a correctly shaped palatal obturator.

**Figure 3 ijerph-18-03219-f003:**
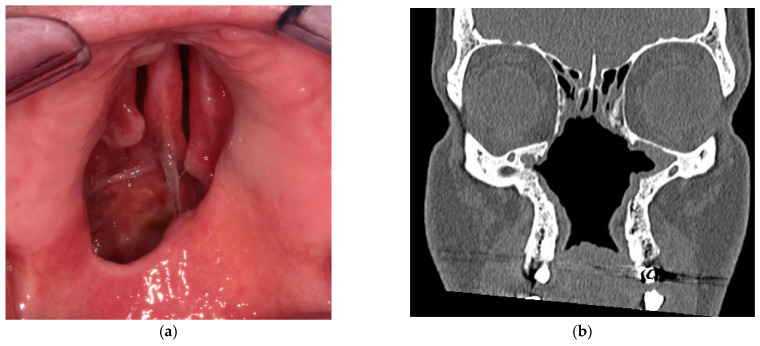
Patient 3: (**a**) Hard and soft palatal perforation at oroscopy. (**b**) A CT scan showing massive erosion of the midline structures, with the destruction of nasal septum, turbinates, and palate.

## Data Availability

The data presented in this study are available on request from the corresponding author. The data are not publicly available due to privacy restrictions.

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
