# Peer review of "Cocaine-Induced Midline Destructive Lesions: A Real Challenge in Oral Rehabilitation"

_ijerph, 2021, doi:10.3390/ijerph18063219_

Round 1

Reviewer 1 Report

Cocaine-induced mid-line destructive lesion are relatively common, the case report presents several limitations: - there are less diagnosis data - there is less description of each case report - there is no follow-up in any cases. 

Author Response

Review: Cocaine-induced mid-line destructive lesion are relatively common, the case report presents several limitations: - there are less diagnosis data - there is less description of each case report - there is no follow-up in any cases. 

Answer: Dear reviewer, thank you for your comments, following which we have added some required information about the case reports and their follow-up in the results section (Page 3, lines 131,132, 137, 138, 147,148).  We are sorry you did not appreciate the appropriateness of this case series, that has been approved by the other reviewers. We hope the revisions will clear your doubts. Thank you very much for the time you are dedicating us.

Reviewer 2 Report

Congratulations to the Authors, now the manuscript is ready for publication.

Author Response

Thank you

Reviewer 3 Report

Dear Authors

I was really pleased to review your manuscript as the topic is interesting and actual.

The manuscript is well organized and materials and methods provide sufficient information and novelty of treatment.

I suggest to accept it

Best regards

Author Response

Thank you

Reviewer 4 Report

The case-report is generally well-written but needs minor revisions related to the conclusion.

I would encourage the authors to split their methdos into subsections such as ethical issues, patient presentation/pre-treatment symptoms, treatment approach etc.

In line 103, please provide the exact age and gender of each of the patients. You just have 3 patients.

In its present format, the conclusion is too long. In my opinion, the conclusion should be "straight to the point". Please rephrase and "simplify" the conclusion.

Author Response

  1. The case-report is generally well-written but needs minor revisions related to the conclusion. I would encourage the authors to split their methdos into subsections such as ethical issues, patient presentation/pre-treatment symptoms, treatment approach etc.

Answer: We thank you for appreciating our revision. We divided the “material and methods” and the “results” section in subheadings.

  1. In line 103, please provide the exact age and gender of each of the patients. You just have 3 patients.

Answer: We thank you for your insight. We modified the section accordingly.

  1. In its present format, the conclusion is too long. In my opinion, the conclusion should be "straight to the point". Please rephrase and "simplify" the conclusion

Answer: We recognize the appropriateness of your claim; we modified the section accordingly.

Round 2

Reviewer 1 Report

This article is  limited to some extent by its design and analysis methods, but analyzes the valuable data and provides useful information that deserve publication in order to contribute to further research in this field.

This manuscript is a resubmission of an earlier submission. The following is a list of the peer review reports and author responses from that submission.

Round 1

Reviewer 1 Report

Interesting observation but limited supporting data. Paper should be re-positioned as a case study due the limited sample size.

Reviewer 2 Report

This is a very interesting and well written manuscript. The importance of this research, appropriateness of topic, methodological design, quality and clarity of writing, makes it of great interest to IJERPH readers and to Academics. I would have only a few suggestions to Authors before publication. Authors should add definition of acronym ANCA (Anti-Neutrophil Cytoplasmic Antibodies): pag 4 in Discussion, line 158. (Instead of pag 6, lines 232-233) About the references, the Authors should check and correct the punctuations and some inaccuracies, as in n.8.

Reviewer 3 Report

I agree completely that the cocaine related destruction of these midline structures in the midface represent a tremendous reconstructive challenge.

I would like to see more in the introduction re: the pathophysiology which is basically necrosis due to drug induced ischemia and mucosal damage to an area that is know to be dependent on small vasculature without much in the way of collateral circulation.  The margins of the lesions are at risk as "watershed areas "of vascular supply thus very prone to further necrosis if injured or inflamed.

I would also like to know if other risk factors for progression of lesions existed...especially cigarette smoking

Reviewer 4 Report

Dear Authors,

The manuscript is not suitable for publication in International Journal of Environmental Research and Public Health in its present format.
Significant revisions or more data are required in the manuscript. Please add more case report specially add CT scan report and try to explain details about your specific findings.